# Antiviral and anti-inflammatory evaluation of herbal extracts: Implications for the management of calf diarrheal diseases

Xi-Rui Xiang, Eun-Seo Lee, Junho Lee, Su Min Kyung, Han Sang Yoo◯*

Department of Infectious Disease, College of Veterinary Medicine, Seoul National University, Seoul, Republic of Korea

* yoohs@snu.ac.kr

## Abstract

Traditional herbal extracts are attracting attention in the context of animal disease control because of their low side effects, diverse bioactive compounds, and low anti-microbial resistance risk. However, the underlying mechanisms remain inadequately understood. To characterize the multifaceted biological activities that underlie their therapeutic potential, this study systematically evaluated the antiviral and complex immunomodulatory properties of three distinct herbal combinations (designated Extracts A, B, and C) *in vitro*. The antiviral activities of the extracts were tested against bovine rotavirus and bovine coronavirus, two major pathogens of neonatal calf diarrhea, and their regulation of inflammatory mediators was assessed in a murine macrophage model (RAW 264.7 cells) stimulated with lipopolysaccharide from *Escherichia coli* by measuring nitric oxide production and the gene expression of inflammation related enzymes. Immunomodulatory effects were investigated by analyzing the gene expression of T helper cell-associated cytokines in both RAW 264.7 cells and bovine peripheral blood mononuclear cells (PBMCs). All three extracts exhibited inhibitory activity against both viruses and reduced the expression of specific inflammatory mediators. Furthermore, the extracts demonstrated complex immunomodulatory effects in bovine PBMCs, with Extract C promoting T helper 1 and T helper 17 responses while suppressing the regulatory T-cell transcription factor Forkhead Box P3. In conclusion, this *in vitro* study demonstrates that these herbal extracts possess antiviral and immunomodulatory potential, providing a basis for future studies to determine their relevance in viral infections associated with calf diarrhea.

## Introduction

Diarrhea is among the leading causes of morbidity and mortality in newborn calves, resulting in substantial economic losses [1,2] and posing a serious threat to the

**Data availability statement:** All relevant data are within the manuscript and its Supporting information files.

**Funding:** This study was supported by SMTECH grant number S3346171, the BK21 FOUR program, and the Research Institute for Veterinary Science, Seoul National University, Republic of Korea. The funders had no role in study design, data collection and analysis, decision to publish, or preparation of the manuscript.

**Competing interests:** The authors have declared that no competing interests exist.

cattle industry across Asia and worldwide [3–5]. Among the diverse etiologies of the disease, including viruses, bacteria and parasites [6–8], bovine rotavirus (BRV) and bovine coronavirus (BCoV) are the predominant viral pathogens of enteritis [9,10]. Both viruses severely damage the integrity of the intestinal epithelium, leading to dehydration, electrolyte imbalance and even death [11–13].

At present, clinical treatment strategies for calf diarrhea rely mainly on supportive care [8], and often involve the use of antibiotics to prevent or control secondary bacterial infections [14,15]. However, the widespread use of antibiotics has exacerbated not only the global crisis of bacterial resistance [16], but also complicated subsequent treatments [5,17]. Importantly, the damage caused by viral infection extends beyond direct cytopathic injury to the gut epithelium, and frequently involves dysregulated inflammatory responses [18], such as excessive inflammatory reactions and disruption of immune homeostasis [19]. In antiviral host defense, the immune response driven by T cells plays a vital role [20,21]. CD4+ T helper (Th) cells represent a significant population [20], and include key components such as Th1, Th2, Th17, and regulatory T-cells (Treg) subsets [22]. The balance among these cells is critical, as each produces cytokines and transcription factors that initiate the immune response [23]. Th1 cells promote cellular immunity and antiviral responses through the secretion of cytokines such as interferon-gamma (IFN-γ), interleukin-2 (IL-2), and tumor necrosis factor-alpha (TNF-α), which are regulated by the transcription factor T-box 21 (TBX21, or T-bet) [24], whereas Th2 cells facilitate humoral immunity through cytokines, including IL-4, IL-5, IL-10 and IL-13, under the control of GATA binding protein 3 (GATA3) [25,26]. Th17 cells, characterized primarily by IL-17 and IL-22, are driven by the transcription factor RAR-related orphan receptor C (RORC), and are involved in mucosal defense and inflammatory pathology [27], whereas Treg cells produce cytokines such as IL-10 and transforming growth factor-beta (TGF-β) to suppress excessive immune activation, with high expression of cytotoxic T-lymphocyte antigen 4 (CTLA-4) and the transcription factor Forkhead Box P3 (Foxp3), to prevent tissue damage [28].

Because viral diarrhea involves both viral infection and dysregulated inflammatory responses, agents that can influence both processes are of particular interest. This concept provides the rationale for evaluating the antiviral and immunomodulatory activities of compound herbal extracts. Natural herbs have attracted increasing attention because of their properties involving multiple components and targets [29,30], allowing them to modulate both viral activity and host immunity. Moreover, herbal materials exert relatively low selective pressure for antimicrobial resistance, a feature consistent with One Health goals of mitigating drug resistance. For example, *Scutellaria baicalensis* [31–33], *Commiphora myrrha* [34,35] and *Nypa fruticans* [36,37] are known for their potent anti-inflammatory effects. *Boswellia serrata* [38–40] and *Gardenia jasminoides* [41]has been confirmed to have broad antiviral activity. Propolis, on the other hand, has significant immunomodulatory functions [42–44]. Based on these properties, three compound herbal extract formulations (Extracts A, B, and C) were prepared and standardized for evaluation.

To characterize their multifaceted biological activities, this study systematically evaluated the antiviral and immunomodulatory properties of these three herbal

extracts *in vitro* using viral infection models and immune cell assays. The objective was to provide foundational evidence supporting further investigation of these formulations in the context of viral infections associated with calf diarrhea.

## Materials and methods

All experiments were carried out in the laboratory facilities of the Research Institute for Veterinary Science, Seoul National University, and were performed by the authors of this study. All animal-related procedures were approved by the IACUC of Seoul National University (SNU-240219-4-1).

### Herbal materials

Extract A, Extract B, and Extract C used in this study were provided by K Pharms Co., Ltd. (Suwon, Korea; formerly Yeskin Co., Ltd.). The compositions were as follows: Extract A was composed of *B. serrata, C. myrrha* and propolis, and Extract B was a single extract of *N. fruticans*. Extract C was a compound of *B. serrata, C. myrrha, S. baicalensis, G. jasminoides*, and propolis. These extracts were selected based on their demonstrated bioactivities in gastrointestinal or inflammatory conditions [31,36,37], and are formulated for veterinary health support. According to the information provided by the manufacturer, these extracts were produced using standard procedures for solvent extraction, although the detailed steps of the extraction process are proprietary and not publicly disclosed. All the extracts were dissolved in DMSO to prepare 200 mg/mL stock solutions, which were stored at −20 °C in the dark. Working concentrations of DMSO were freshly prepared immediately before experimentation. The final DMSO concentration did not exceed 0.1% (v/v), and vehicle controls were included accordingly.

### Cells and virus culture

**Cell lines and virus.** Madin-Darby Bovine Kidney (MDBK) cells, TF104 cells (a clonal derivative of MA-104 monkey kidney cells), and RAW 264.7 mouse macrophages, all of which were maintained as laboratory stocks, were used in this study. Cells were used at passages 5–10 after thawing. All cells were cultured in Dulbecco's modified Eagle's medium (DMEM; Gibco, USA) supplemented with 10% heat-inactivated fetal bovine serum (FBS; Gibco) and 100 µg/mL streptomycin and 100 U/mL penicillin (Gibco), and incubated at 37 °C and 5% $CO_2$. BCoV (strain number: VR1500040) was obtained from the Animal and Plant Quarantine Agency (Gimchen, Korea), and BRV was isolated from lyophilized commercial vaccines provided by Choong Ang Vaccine Laboratories Co., Ltd. (Daejeon, Korea). BCoV was propagated in MDBK cells, and BRV was propagated in TF104 cells. Infectious titers (50% tissue culture infectious dose, $TCID_{50}$) were determined by the Reed–Muench method [45], yielding titers of $1 \times 10^{5.62}$ $TCID_{50}$/mL for BCoV and $1 \times 10^{2.86}$ $TCID_{50}$/mL for BRV.

**Isolation and culture of bovine PBMCs.** Heparinized jugular venous blood was collected from healthy cattle maintained at an experimental farm of Seoul National University. Bovine peripheral blood mononuclear cells (PBMCs) were isolated by centrifugation at $800 \times g$ for 15 min through Leucosep tubes (Greiner Bio-One, Austria). The PBMC layer was harvested, washed with Dulbecco's phosphate-buffered saline (DPBS; Gibco), and treated with red blood cell lysis buffer (Roche Diagnostics, Germany). The cells were re-suspended in RPMI 1640 medium (Gibco), supplemented with 10% heat-inactivated FBS and penicillin–streptomycin (100 U/mL; 100 µg/mL), and then cultivated at 37°C with 5% $CO_2$ until use.

### Cytotoxicity assay

MDBK, TF104, RAW 264.7 cells and bovine PBMCs were seeded in 96-well plates at a density of $1–4 \times 10^4$ cells/well, respectively. The plates were incubated overnight at 37°C to allow for cell attachment. For all experimental conditions, including the negative control (NC), positive control (PC), and extract-treated groups, assays were performed in

quadruplicate wells. Subsequently, the culture medium was replaced with fresh medium containing each extract at concentrations of 6.25, 12.5, 25, 50, 100 and 200 µg/mL. The NC group was treated with 0.1% DMSO. For bovine PBMCs, 1 µg/mL lipopolysaccharide (LPS) from *Escherichia coli* (Sigma-Aldrich, USA) was added to the PC group, which was included as a positive reference for subsequent immunoactivity assays. In accordance with the cell growth characteristics, the viral host cells MDBK and TF104 were treated for 48 h and 96 h, respectively, to determine the optimal time points for evaluating the viral CPE. The immune cells, RAW 264.7 cells and bovine PBMCs, were treated for 24 h and 48 h, respectively, representing the optimal duration for assessing their immunomodulatory responses while maintaining good cell viability.

Adherent cells were processed with MTT reagent (5 mg/mL stock; Thermo Fisher Scientific, USA) for 4 h according to the manufacturer's instructions. PBMCs received 10 µL of WST-1 (Abcam, UK) per well. The absorbance was measured on a microplate reader at 570 nm for MTT and 450 nm for WST-1.

## RNA extraction and reverse transcription

Total RNA was extracted using TRIzol Reagent (Thermo Fisher Scientific). After chloroform phase separation, the RNA was precipitated with isopropanol, washed with 75% ethanol, and finally resuspended in DEPC-treated water. cDNA was synthesized from total RNA using a High-Capacity cDNA Reverse Transcription Kit (Applied Biosystems, USA) following the manufacturer's protocol and then stored at −20 °C until use.

## Antiviral effect on BCoV and BRV

MDBK ($2 \times 10^4$ cells/well) and TF104 cells ($2 \times 10^4$ cells/well) were seeded in 6-well plates overnight and inoculated with either BCoV (MOI = 0.1) or BRV (MOI = 0.01) for 1 h. A negative control (NC; vehicle-matched 0.1% DMSO) and a virus-only positive control (PC) were included. After adsorption, the inoculum was replaced with medium containing Extracts A–C at 6.25, 12.5 or 25 µg/mL. The cultures were continued for 48 h in BCoV or 96 h in BRV until CPE appeared. Afterward, the cells were harvested for RNA extraction and subsequent RT-qPCR analysis.

Relative viral gene expression was quantified by RT-qPCR targeting the BCoV M gene and the BRV VP1 gene on a Rotor-Gene Q (QIAGEN, Germany). Reactions contained 10 µL of EzAmp™ qPCR 2 × Master Mix (SYBR Green; ELPIS-BIOTECH, South Korea), 2 µL of cDNA template, and 10 pmol of each primer. Cycling: 95 °C for 3 min; 35 cycles of 95 °C for 15 s and 55–60°C for 55 s. The primers that were designed in this study were generated using Primer-BLAST (NCBI) with default parameters and targeting coding sequences obtained from GenBank. Primer specificity was confirmed by *in silico* BLAST analysis and by melting curves exhibiting a single peak in qPCR. And then synthesized by Macrogen Inc. (Seoul, Korea), and the sequences are shown in Table 1. The expression was normalized to that of β-actin and analyzed using the $2^{-\Delta\Delta Ct}$ method [46]. The antiviral experiments were independently repeated three times.

**Table 1. Sequences of primers used for bovine coronavirus and bovine rotavirus viral RNA by RT-qPCR.**

| Targets | Primer names | Sequence 5′–3′ | Accession Number | References |
|---|---|---|---|---|
| Viral RNA | BCoV-F | CTGGAAGTTGGTGGAGTT | KX897162.1 | [46] |
| | BCoV-R | ATTATCGGCCTAACATACATC | | |
| | β-actin-MDBK-F | CTGCGGCATTCACGAAAC | NM_173979.3 | This study |
| | β-actin-MDBK-R | GCAGTGATCTCTTTCTGCATC | | |
| | BRV-F | TTCCGCGCTGCTAACTATGT | PV026141.1 | This study |
| | BRV-R | TGGCAAACTAGCCTCAGTGG | | |
| | β-actin-TF104-F | CGGGACCTGACTGACTACCT | NM_001033084.1 | This study |
| | β-actin-TF104-R | R:CTCCTGCTCGAAGTCCAGG | | |

## Regulation of Inflammatory Mediators in RAW264.7 Cells

RAW 264.7 cells ($1 \times 10^4$ cells/well; 96-well plates) were used to evaluate the regulation of inflammatory mediators. Prior to treatment, the cells were incubated overnight to allow for adherence. In an extract-only check, cells treated with Extracts A–C (6.25–25 µg/mL; vehicle 0.1% DMSO) for 24 h showed no increase in nitric oxide (NO) by the Griess reagent system (Promega, USA) at 540 nm (data not shown). For the main assay, cells were pretreated with different concentrations (6.25, 12.5 and 25 µg/mL) of the Extracts A–C for 4 h, and then coincubated with LPS-*E.coli* at 1 µg/mL for 24 h. A negative control (NC; vehicle only, no LPS) and a positive control (PC; LPS only) were also included. The cell supernatant was collected to quantify production of nitric oxide (NO) by the Griess Reagent System. The cell pellets were harvested for total RNA extraction. The mRNA expression levels of two representative proinflammatory genes, inducible nitric oxide synthase (iNOS) and cyclooxygenase-2 (COX-2), were measured by RT-qPCR with β-actin as the housekeeping gene, according to the method described above. The primers used are listed in Table 2. The experiments were independently repeated three times.

## Immunomodulatory effects on RAW 264.7 cells and bovine PBMCs

RAW 264.7 macrophages and bovine PBMCs were seeded in 6-well plates at $5 \times 10^5$ and $2$–$3 \times 10^5$ cells/well, respectively, and incubated overnight to allow for cell attachment. The cells were treated with different concentrations (6.25, 12.5 and 25 µg/mL) of extracts, LPS-*E. coli* (1 µg/mL, positive control), or 0.1%DMSO (negative control) for 4, 8 and 24 h in RAW 264.7 cells or 4, 8, 24 and 48 h in bovine PBMCs. At each time point, total RNA was extracted, and RT–qPCR was performed as previously described, while "relative" fold changes in cytokine expression levels were compared with those in the negative control group by the $2^{-\Delta\Delta Ct}$ method. The targets included immunomodulatory cytokines and transcription factors, as listed in Table 3. The experiments were independently repeated three times.

## Statistical analysis

All experiments were repeated a minimum of three times, and the data are presented as the mean ± standard deviation (SD). The information was analyzed using IBM SPSS Statistics 26, and figures were generated with GraphPad Prism 10.0.0. Comparisons among multiple groups were conducted using one-way analysis of variance (ANOVA), followed by Dunnett's post-hoc test for pairwise comparisons between each extract-treated group and the corresponding control group. A p-value of less than 0.05 was considered to indicate statistical significance (*$p < 0.05$, **$p < 0.01$, ***$p < 0.001$).

# Results

## Cytotoxicity across four cell types

To systematically evaluate cellular safety, four cell types were treated with Extracts A–C, as shown in Fig 1, and their viability is expressed relative to that of the vehicle control (100%). In viral host MDBK cells, only Extract A at 200 µg/

**Table 2. Sequences of primer for the detection of inflammation-related genes in RAW 264.7 cells by RT–qPCR.**

| Targets | Primer names | Sequence 5′–3′ | Accession Number | References |
|---|---|---|---|---|
| Inflammatory cytokines | iNOS-F | GAGCCACAGTCCTCTTTGCT | NM_001429940.1 | This study |
| | iNOS-R | CAACCTTGGTGTTGAAGGCG | | |
| | COX-2-F | TGCCCGACACCTTCAACATT | XM_028775752.1 | This study |
| | COX-2-R | AGAAGCGTTTGCGGTACTC | | |
| | β-actin-RAW-F | GCTCCGGCATGTGCAAAG | NM_007393.5 | This study |
| | β-actin-RAW-R | CCTTCTGACCCATTCCCACC | | |

**Table 3. Primer sequences used for the quantification of gene in RAW264.7 cells and bovine peripheral blood mononuclear cells by RT-qPCR.**

| Targets | Primer names | Sequence 5'–3' | Accession Number | References |
|---|---|---|---|---|
| Genes in RAW264.7 | TNF-α-F | TTCTATGGCCCAGACCCTC | NM_013693.3 | [47] |
| | TNF-α-R | ACTTGGTGGTTTGCTACG | | |
| | IL-5-F | CCATGAGCACAGTGGTGAAAG | NM_010558.1 | This study |
| | IL-5-R | GACAGGAAGCCTCATCGTCTC | | |
| | IL-6-F | TACTCGGCAAACCTAGTGCG | NM_031168.2 | [48] |
| | IL-6-R | GTGTCCCAACATTCATATTGTCAGT | | |
| | IL-10-F | ACCTGGTAGAAGTGATGCCC | NM 010548.2 M | This study |
| | IL-10-R | TGTAGACACCTTGGTCTTGGA | | |
| | IL-17-F | CTCAGACTACCTCAACCGTTCC | NM_001106897.1 | This study |
| | IL-17-R | GTGCCTCCCAGATCACAGAAG | | |
| | IL-21-F | CAAGCCATCAAACCCTGGAAAC | NM_021782.3 | This study |
| | IL-21-R | TTCTCATACGAATCACAGGAGGG | | |
| | β-actin-F | CATTGCTGACAGGATGCAGAAGG | NM_007393.1 | This study |
| | β-actin-R | TGCTGGAAGGTGGACAGTGAGG | | |
| Genes in PBMCs | IFN-γ-P-F | AATTCCGGTGGATGATCTG | NM_174086.1 | This study |
| | IFN-γ-P-R | GATTCTGACTTCTCTTCCGC | | |
| | T-bet-P-F | ATTACCGGATGTATGTGGAC | NM_001192140 | This study |
| | T-bet-P-R | GCTTTAGTTTCCCGAATGAC | | |
| | IL-4-P-F | GAATTGAGCTTAGGCGTATC | NM_173921.2 | This study |
| | IL-4-P-R | CTTCATTCACAGAACAGGTC | | |
| | GATA3-P-F | CTACCACAAGATGAACGGAC | XM_005214126.5 | This study |
| | GATA3-P-R | AGAGTCGTAGTTGTGGTTTG | | |
| | IL17-P-F | TATGTCACTGCTACTGCTTC | NM_001008412.2 | This study |
| | IL17-P-R | GTTTAGGTTGACCCTCACAT | | |
| | RORC-P-F | CACAGAGACATCACCGAG | NM_001083451 | This study |
| | RORC-P-R | TAGTGGATCCCAGATGACTT | | |
| | CTLA4-P-F | ACTTGGTGGACATCTAGGAC | NM_174297.1 | This study |
| | CTLA4-P-R | GGTCACATTCATCCCTTTAG | | |
| | Foxp3-P-F | TCTATCACTGGTTTACACGC | NM_001045933.1 | This study |
| | Foxp3-P-R | TTGCGGAACTCAAACTCAT | | |
| | β-actin-P-F | GATATTGCTGCGCTCGTGGT | NM_173979 | This study |
| | β-actin-P-R | TACGAGTCCTTCTGGCCCAT | | |

mL and Extract C at 100 and 200 μg/mL significantly reduced cell viability to 75.58%, 64.24% and 65.23%, respectively (p < 0.001), whereas the other concentrations did not show obvious cytotoxicity. Treatment of TF104 cells for 96 h resulted in a dose-dependent loss of viability, with 100–200 μg/mL reducing viability to ≤50%. In immune cells, 200 μg/mL Extracts A and C decreased the viability of RAW 264.7 cells to 12.05% and 21.72%, respectively (p < 0.001). In contrast, after treatment with bovine PBMCs for 48 h, the three extracts exhibited no cytotoxicity, and the high concentration treatment significantly increased cell growth. At 200 μg/mL, Extract C increased the cell viability to 231.97% of that of the control group (p < 0.001). Similarly, the PC group showed no significant cytotoxicity to bovine PBMCs. Therefore, based on these findings, noncytotoxic concentrations of 6.25, 12.5, and 25 μg/mL for Extracts A, B, and C, respectively, were selected for all subsequent experiments.

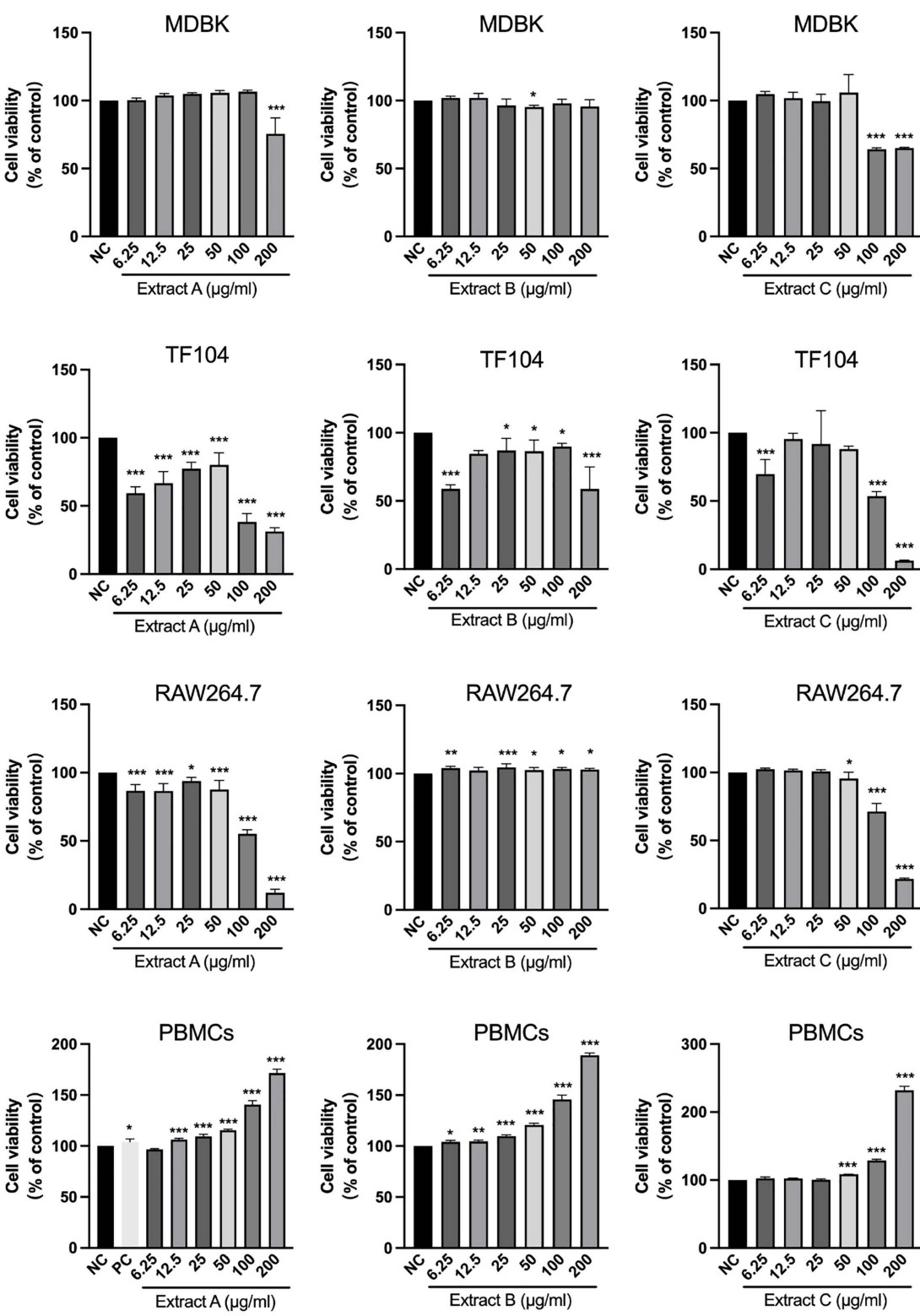

**Fig 1. The extracts have differential cytotoxic effects on different cell types.** MDBK (48 **h)**, TF104 (96 **h)**, RAW264.7 (24 **h)**, and bovine PBMCs (48 h) cells were treated with graded concentrations (6.25–200 μg/mL) of Extract A, B or **C.** The NC group was treated with vehicle (0.1% DMSO), and the PC group was treated with 1 μg/mL LPS. The data are presented as the mean±SD (n=4) and are expressed as a percentage of the NC group. Statistical significance relative to the NC group was determined by one-way ANOVA with Dunnett's post-hoc test (*p<0.05, **p<0.01, ***p<0.001).

## Antiviral effects on BCoV and BRV

As shown in Fig 2, three concentrations (6.25, 12.5, and 25 µg/mL) of all the extracts significantly inhibited BCoV. Extract A exhibited the strongest suppression at 25 µg/mL (p < 0.001), whereas Extracts B and C showed greater inhibition at 6.25–12.5 µg/mL than at 25 µg/mL. The extracts presented distinct patterns of activity against BRV. The effect of Extract A was strongly dose dependent, with greater inhibition observed at higher doses. At all tested concentrations, Extract B reduced viral replication. However, Extract C showed a biphasic pattern, with an inhibitory effect at the low concentration and a significant increase at the highest concentration of 25 µg/mL (p < 0.001).

## Herbal extracts reduce NO and COX-2 but differentially modulate iNOS

The regulation of inflammatory mediators by the extracts were evaluated in LPS-stimulated RAW 264.7 cells (Fig 3). At the transcript level, COX-2 mRNA expression was significantly suppressed by all the extracts. The inhibitory effects of Extracts A and C increased with increasing concentration, as Extract B significantly reduced COX-2 expression at all doses (p < 0.001). However, with respect to iNOS expression, the results revealed that compared with the positive control, Extract B significantly reduced iNOS expression only at 6.25 µg/mL (p < 0.01). Additionally, NO production decreased

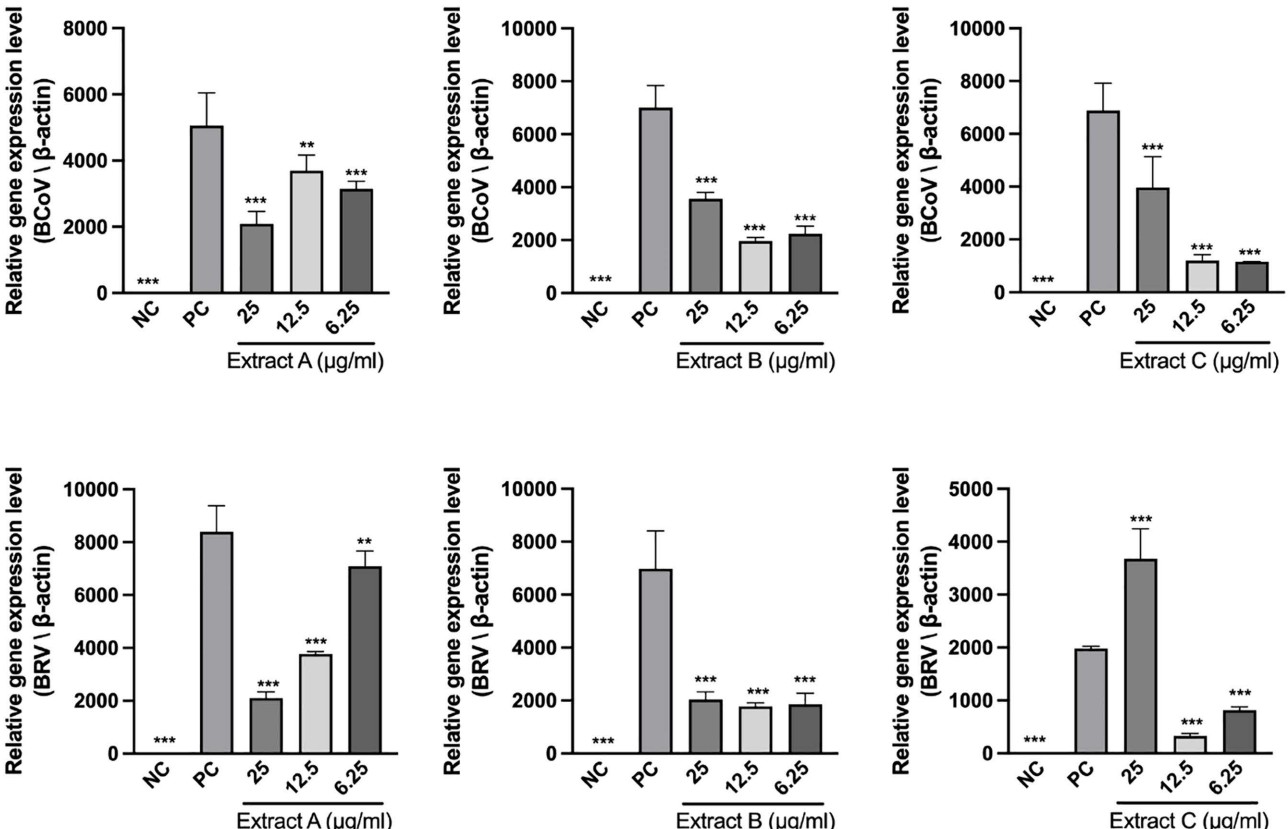

**Fig 2. Antiviral effects of the extracts on BCoV and BRV.** MDBK and TF104 cells were infected with BCoV (MOI = 0.1) and BRV (MOI = 0.01), respectively, and then treated with graded concentrations (6.25–25 µg/mL) of each extract for 48 h or 96 **h.** The relative mRNA expression of the BCoV M gene and the BRV VP1 gene was quantified by RT-qPCR. NC represents uninfected cells. Data are presented as the mean ± SD (n = 3), and statistical significance relative to the virus control (PC; treated with 0.1% DMSO) was determined by one-way ANOVA with Dunnett's post-hoc test (**p < 0.01, ***p < 0.001).

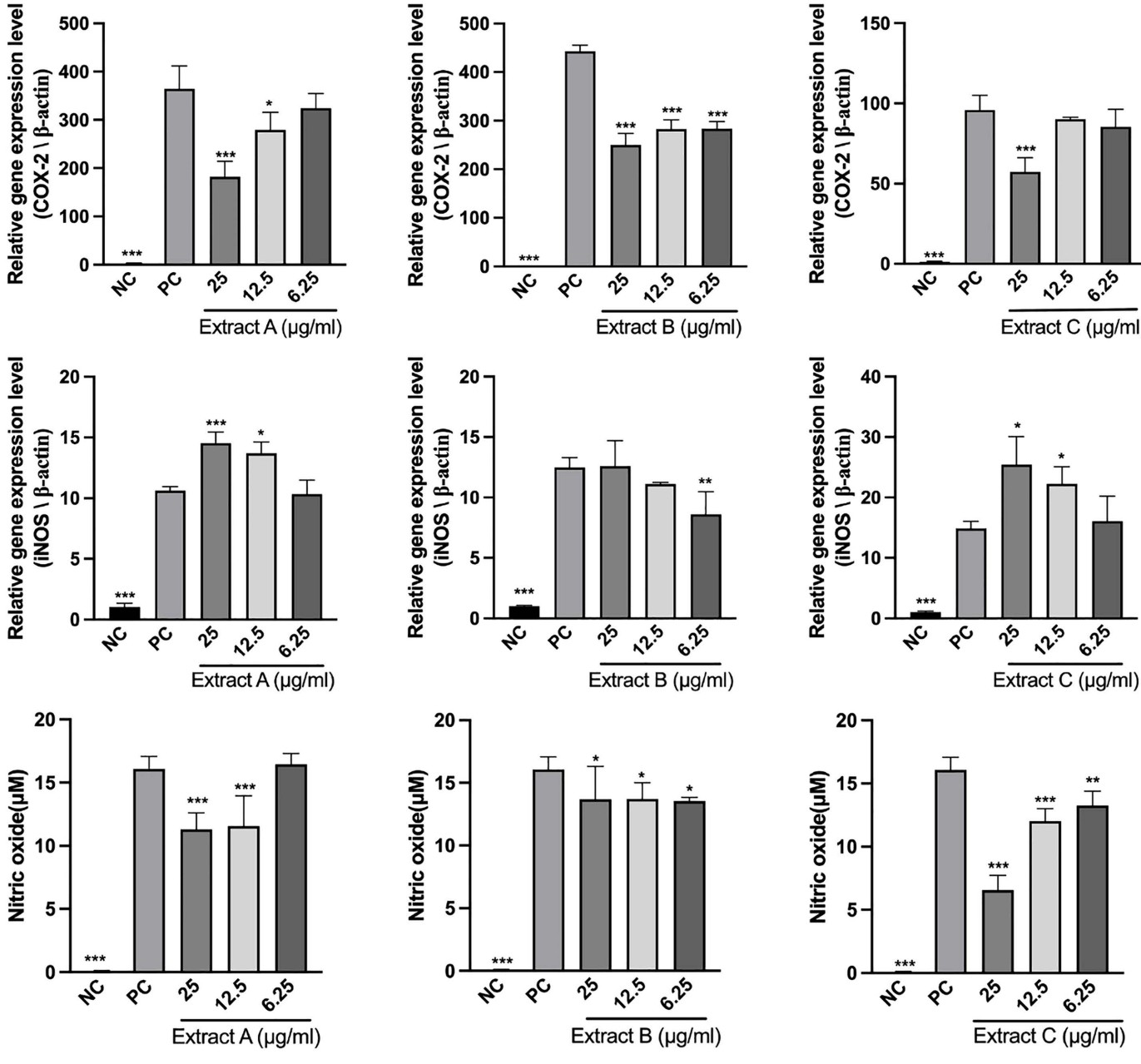

**Fig 3. Regulation of inflammatory mediators by the extracts in LPS-stimulated RAW 264.7 macrophages.** RAW 264.7 cells were treated with 6.25, 12.5 and 25 μg/mL of each extract for 4 h, followed by stimulation with 1 μg/mL LPS-*E. coli* for 24 **h.** The relative mRNA expression levels of COX-2 and iNOS were determined by RT–qPCR and normalized to that of β-actin. The concentration of nitrite in the cell supernatant was measured using a Griess reagent system. The NC group was left untreated, while the PC group was stimulated with only LPS. The data are presented as the mean ± SD (n = 3). Statistical significance relative to the PC group was determined by one-way ANOVA with Dunnett's post-hoc test (*p < 0.05, **p < 0.01, ***p < 0.001).

with all three extracts. It was reduced by Extract C at concentrations ranging from 6.25–25 μg/mL in a dose-dependent manner. At 12.5 and 25 μg/mL, Extract A reduced NO levels (p < 0.001), whereas Extract B did so at all the tested doses (p < 0.05).

## Extracts induce differential expression of immunomodulatory genes in RAW 264.7 cells

All three extracts induced dynamic and distinct gene expression profiles over a 24 hours period. As shown in Fig 4, 25 µg/mL Extract A caused significant induction of TNF-α expression at 24 h (p<0.001). This extract also induced peak expression of IL-6 and IL-17 at 8 h. For Extract B (Fig 5), treatment significantly increased TNF-α expression at 8 hours. In contrast, Extract C (Fig 6) at 6.25 µg/mL significantly induced IL-6 expression at 24 h. In a separate control experiment (S1 Fig), stimulation with LPS-*E. coli* caused a sustained increase in the expression of most proinflammatory genes, confirming the responsiveness of the assay system.

## Expression of T cell-associated genes in bovine PBMCs in response to extracts

As shown in Fig 7, all three concentrations of Extract A caused initial upregulation of CTLA4 at 4 h. The 6.25 µg/mL treatment subsequently increased the TBX21 level at 8 h (p<0.001), and the 25 µg/mL treatment induced an increase in the IFN-γ level at the final 48-hour time point. Treatment with Extract B at 25 µg/mL resulted in a significant initial upregulation of IFN-γ, TBX21, IL-4, IL-17 and CTLA4 expression (Fig 8; p<0.01). However, all concentrations of Extract B suppressed GATA3 expression throughout the entire time course. In cells treated with Extract C, 25 µg/mL led to significant increases in TBX21 and IL-4 expression at 4 h, whereas 12.5 µg/mL caused increases in IFN-γ and IL-17 expression at later stages (p<0.001), with recovery of IL-4 and CTLA4 expression observed at 24 h across all concentrations (Fig 9). After LPS-*E.*

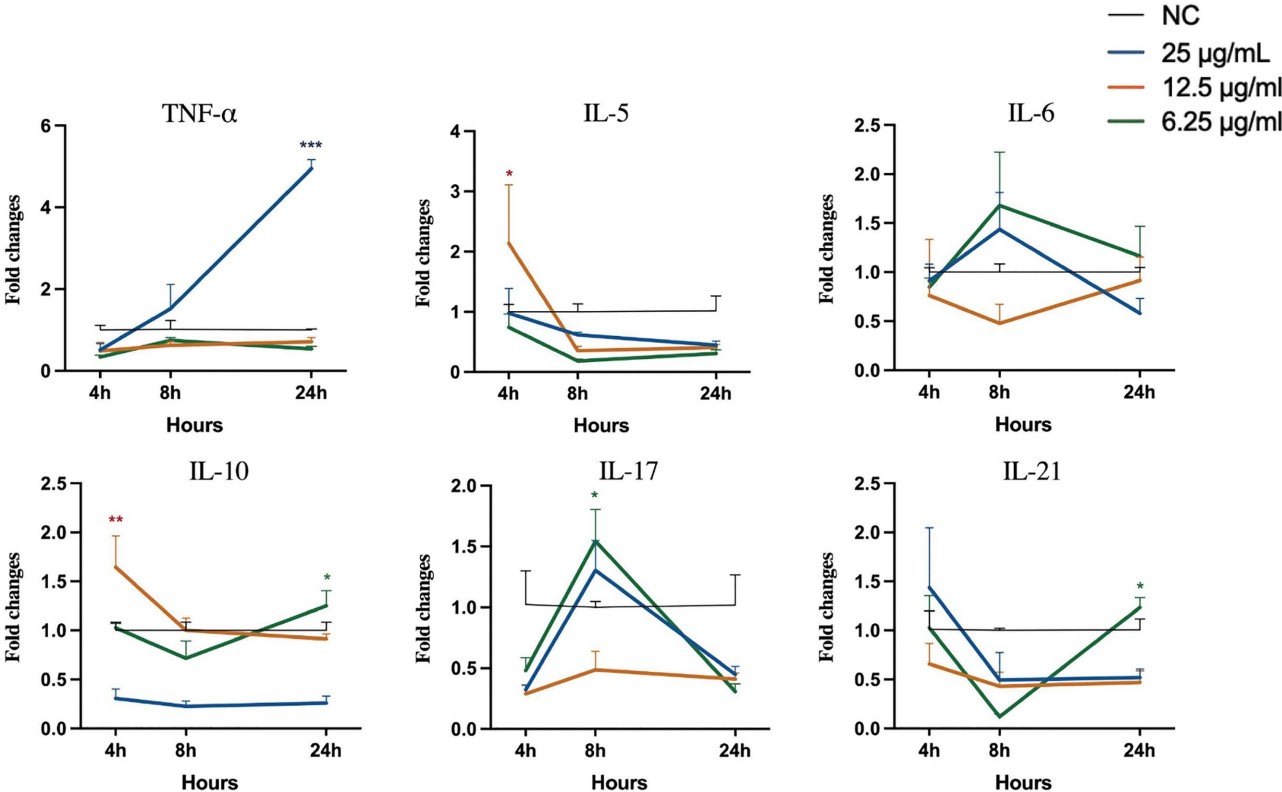

**Fig 4. Analysis of cytokine gene expression in RAW 264.7 cells stimulated with Extract A.** Asterisks (*p<0.05, **p<0.01, ***p<0.001) indicate that extract-induced gene expression is significantly different from that in the negative control group at the same time points. Statistical significance was evaluated using one-way ANOVA with Dunnett's post-hoc test.

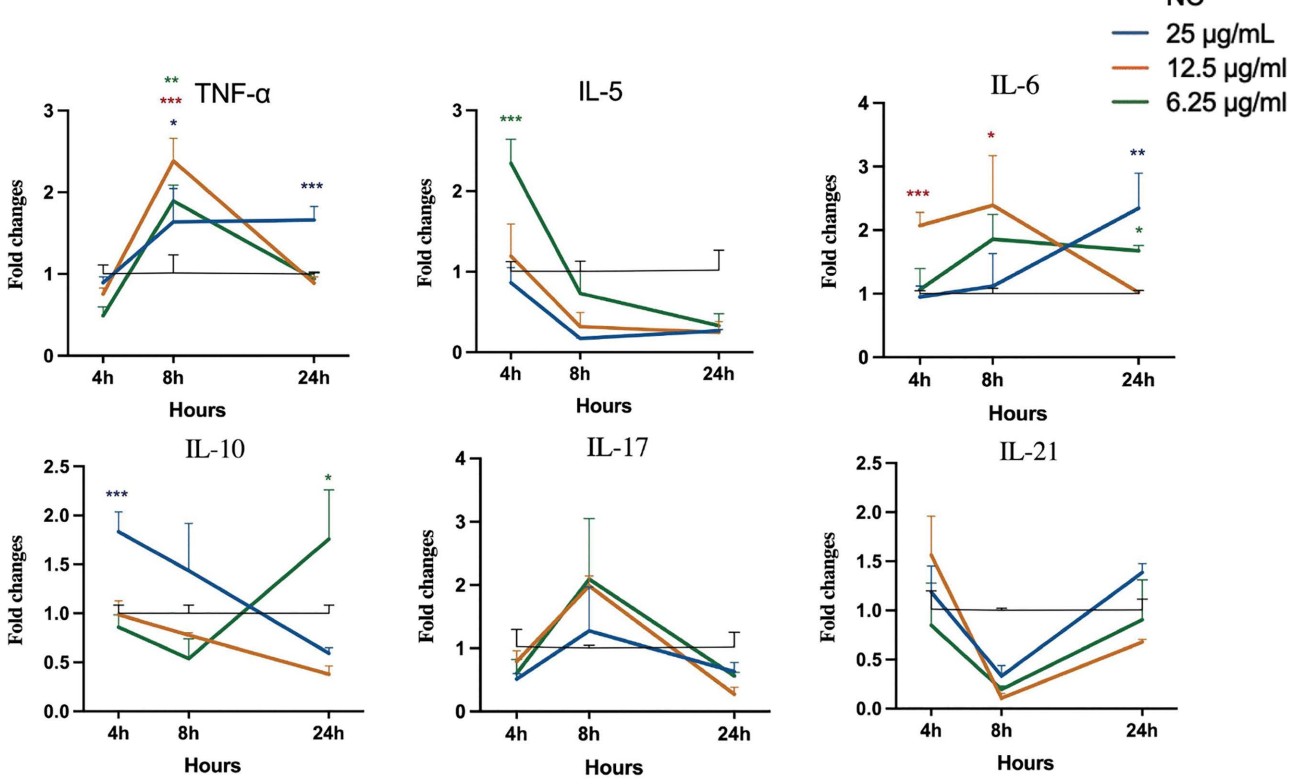

**Fig 5. Analysis of cytokine gene expression in RAW 264.7 cells stimulated with Extract B.** Asterisks (*p < 0.05, **p < 0.01, ***p < 0.001) indicate that extract-induced gene expression is significantly different from that in the negative control group at the same time points. Statistical significance was evaluated using one-way ANOVA with Dunnett's post-hoc test.

*coli* stimulation, the mRNA expression of IFN-γ, TBX21, IL-17, and CTLA4 increased, peaking at 24 and 48 h, which established a benchmark for a proinflammatory response (S2 Fig).

## Discussion

Neonatal calf diarrhea remains a persistent challenge to the global cattle industry owing to its high incidence rate and the limitations of available therapies [8,9,49]. This multifactorial disease involves both infectious and noninfectious factors, with a high probability of mixed infection by multiple pathogens [50,51]. As potential and valuable medicinal resources with intrinsic multitarget properties, natural herbs are gaining increasing attention in modern veterinary clinical research. [52,53]. Although previous studies have reported the efficacy of certain individual herbs [39,43], the combined antiviral and immunomodulatory effects of combinations of these herbs against BCoV and BRV have not been systematically studied.

In this study, all three herbal extracts clearly inhibited calf diarrhea viruses (BCoV and BRV) at concentrations that did not show detectable cytotoxicity in the cell models used. Notably, Extracts B and C showed strong antiviral effects even at relatively low concentration and the activity of Extract C is consistent with the reported broad-spectrum antiviral properties of its constituents *B. serrata* and *S. baicalensis* [38,54]. However, the antiviral effect reversed at higher concentrations, especially for Extract C. Possible mechanisms include a hormetic response [55], and the noncompetitive interactions of herbal compounds with host entry receptors at defined concentrations [56].

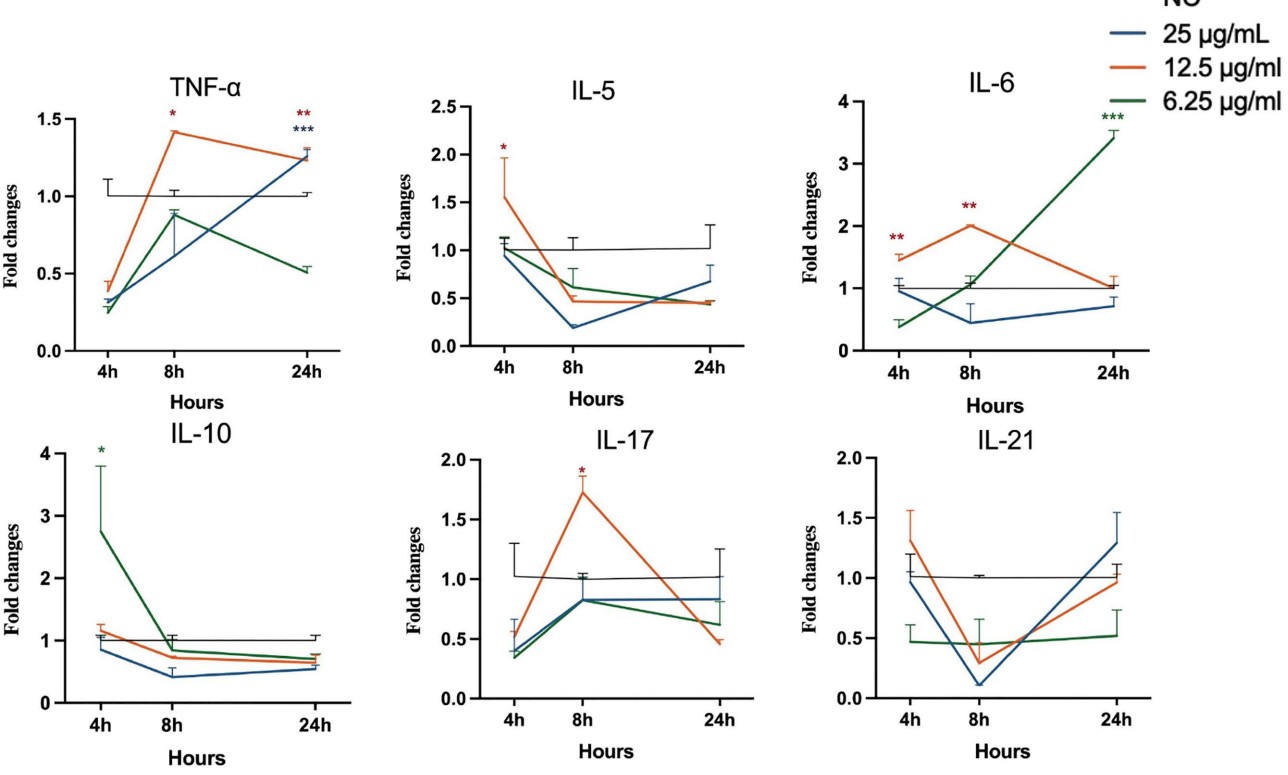

**Fig 6. Analysis of cytokine gene expression in RAW 264.7 cells stimulated with Extract C.** Asterisks (*p < 0.05, **p < 0.01, ***p < 0.001) indicate that extract-induced gene expression is significantly different from that in the negative control group at the same time points. Statistical significance was evaluated using one-way ANOVA with Dunnett's post-hoc test.

The distinct antiviral profiles of Extract A and Extract C (Fig 2) suggest a potential divergence in their mechanisms of action, likely driven by the differential herbal composition of the two formulations. Specifically, Extract A demonstrated efficacy against BCoV, whereas Extract C displayed prominent activity against BRV. These patterns likely stem from the differential susceptibility of each viral pathogen to specific herbal constituents, such as the specific contribution of *S. baicalensis* and *G. jasminoides* in Extract C. With respect to enveloped BCoV, inhibitory effects may involve interference with viral entry steps. For example, *C.myrrha* may mediate the modulation of host membrane mediators in influenza virus [57], and the blockade of β-coronavirus spike glycoprotein interactions as reported for SARS-CoV-2 [58]. Although these interactions at the entry level were not directly tested in this study, their relevance to BCoV inhibition warrants further investigation. With respect to nonenveloped BRV, intracellular pathways have been implicated for baicalin from *S. baicalensis* via the p-JNK–PDK1–AKT–SIK2 pathway [59]and for genipin from *G. jasminoides* that exhibits anti-rotavirus activity [41]. Given the complex nature of herb formulations, such interactions may exhibit antagonistic or nonlinear behavior under specific combinations or concentrations. This may explain the anomalous reduction in inhibitory activity observed at high concentrations of Extract C. Similar concentration-dependent antagonism has been observed for other herbal compounds [60], suggesting that rational formulation and dose optimization are essential for future studies.

In the LPS-induced RAW 264.7 macrophage inflammation model, the herbal extracts attenuated inflammatory responses, as reflected by decreased COX-2 expression and suppressed NO production. In particular, the inhibitory effect of Extract C was dose dependent, which is consistent with the report that *B. serrata* [40] and *C. myrrha* [35] are potent anti-inflammatory herbs. LPS-*E. coli* induces an inflammatory response mainly by activating the NF-κB signaling pathway,

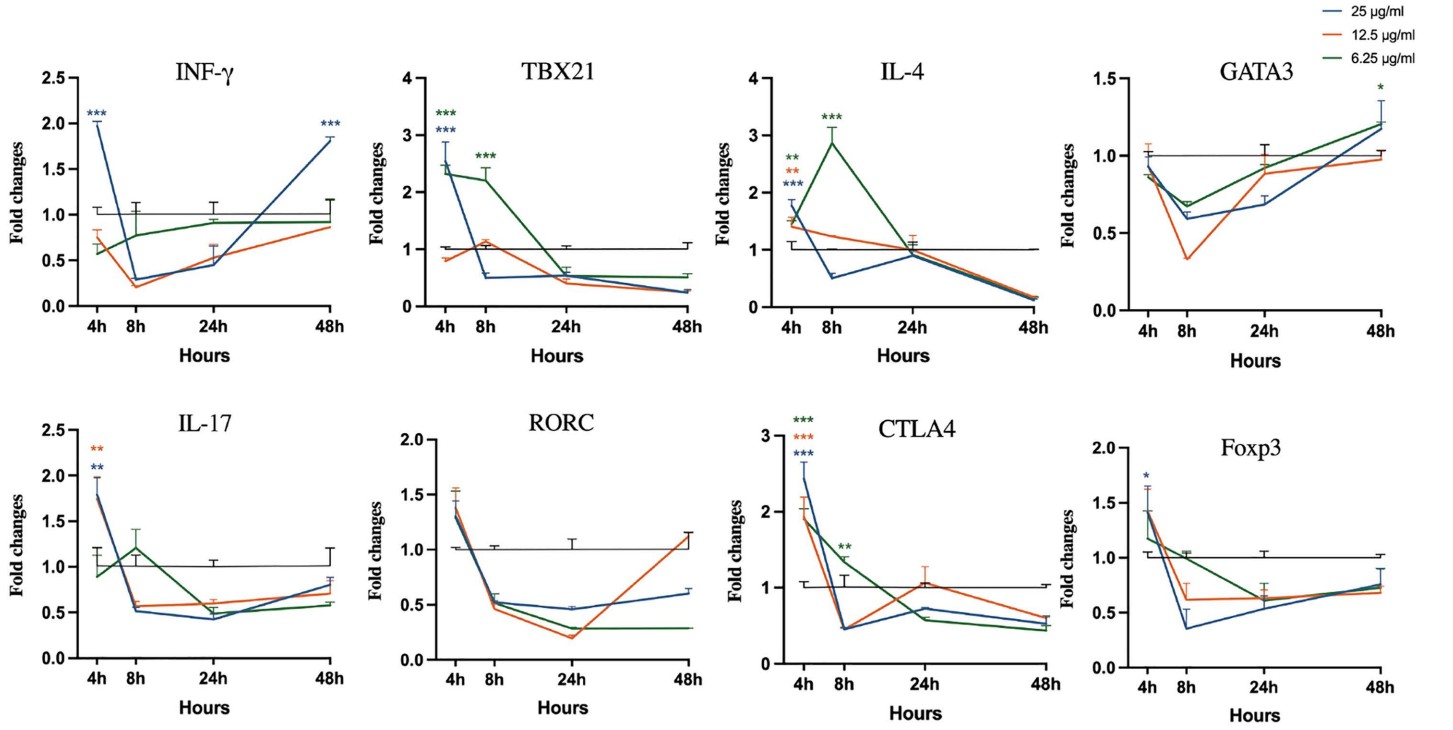

**Fig 7. Analysis of cytokine and transcription factor gene expression in bovine peripheral blood mononuclear cells stimulated with Extract A.** Statistical significance relative to the negative control at each time point was evaluated using one-way ANOVA with Dunnett's post-hoc test (*$p < 0.05$, **$p < 0.01$, ***$p < 0.001$).

which also plays a key regulatory role in intestinal inflammatory conditions such as inflammatory bowel disease [61,62]. Studies have indicated that compounds such as *S. baicalensis* can exert anti-inflammatory effects through the NF-κB pathway, signaling the downregulation of COX-2 expression [63]. Although NF-κB pathway activation was not directly measured in this study, the observed reduction of COX-2 may suggest potential upstream interference with this pathway.

Moreover, the extracts also clearly suppressed NO production, suggesting their potential to block the downstream inflammatory responses. However, their effects on iNOS were more complex. Although iNOS is a direct target of NF-κB [64], its transcription is coregulated by multiple proinflammatory cytokines, including TNF-α, IFN-γ, and IL-6 [62]. Notably, in unstimulated RAW 264.7 cells, the extracts induced TNF-α and IL-6 expression. Therefore, predominant constituents may attempt to downregulate iNOS expression via NF-κB, whereas other components may be counteracted by cytokine induction, suggesting that it may have pathway-selective or multidirectional regulatory effects. Similar discrepancies between iNOS mRNA levels and NO production have been described in research on various natural compounds [65,66]. Although the direct regulatory mechanism of iNOS by the extract remains to be elucidated, the effective inhibition of COX-2 and NO by the extract supports its potential in inflammation-related disease models [67], which warrants further investigation and optimization in the future.

We used two immune cell models, RAW264.7 macrophages and bovine PBMCs, to evaluate the immunomodulatory effects of the herbal extracts in a comparative manner. As a widely used macrophage line [68], RAW264.7 cells provided a baseline for responses associated with macrophages and served as a reference for changes in PBMCs. Even in the absence of external stimulation, the extracts altered the expression of several cytokines, suggesting that certain herbal components may regulate macrophage phenotypes under basal conditions. However, RAW 264.7 cells are not a major

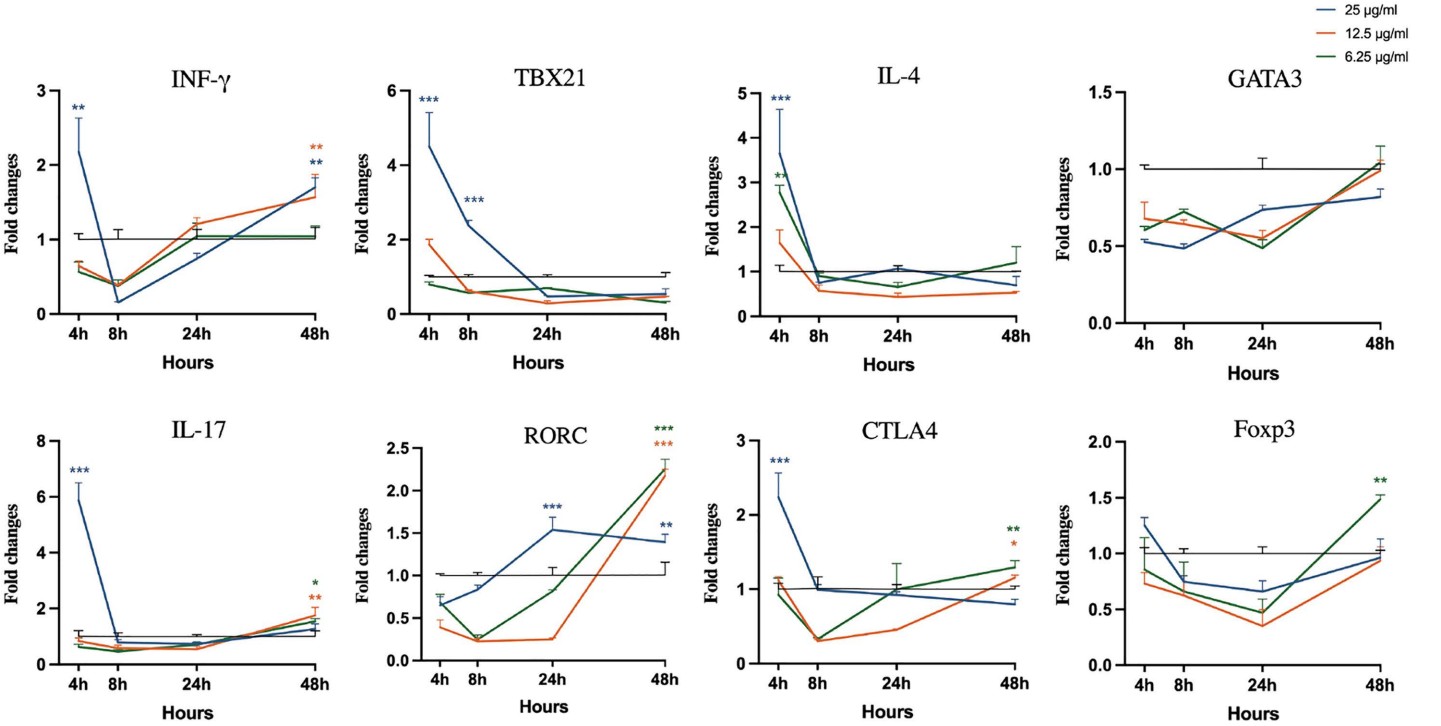

**Fig 8. Analysis of cytokine and transcription factor gene expression in bovine peripheral blood mononuclear cells stimulated with Extract B.** Statistical significance relative to the negative control at each time point was evaluated using one-way ANOVA with Dunnett's post-hoc test (*p < 0.05, **p < 0.01, ***p < 0.001).

source of specific cytokines, such as IL-5 or IL-21. Therefore, only limited changes were observed. In contrast, the PBMC model contains many immune cell types. It showed clearer and more diverse responses [68]. The levels of several cytokines and transcription factors, such as Th1, Th2, Th17, and Treg, changed significantly after extract treatment [24]. These findings suggest that the extracts may have broad immunomodulatory potential in complex immune systems.

As introduced earlier, the dynamic balance immune responses mediated by T helper cells is key to immune regulation. Extract C induced coordinated shifts across makers associated with Th1, Th2, Th17, and Treg cells, with early activation of transcriptional programs related to Th1 and Th2 followed by later enhancement of Th17 markers and concurrent suppression of factors linked to Treg cells. These results suggest that the effects of the extracts can modulate T cell differentiation depending on the treatment conditions. Such multidirectional response is often seen in natural products that work through multiple biological targets. For example, propolis has been reported to activate Th1- and Th2-associated cytokines [69]; *B. serrata* has been shown to downregulate TNF-α, IL-6, and IL-17, as well as T cell activation markers such as CD4 and CD8 [70]; and reports concerning *S. baicalensis* have indicated that natural components can alleviate inflammatory responses via the NF-κB and NLRP3 pathways [71]. Collectively, these findings support the potential of compound extracts to coordinate immune responses by modulating representative markers associated with T cell differentiation.

## Conclusion

In conclusion, this *in vitro* study demonstrates that the three tested herbal extracts possess significant antiviral activity against BCoV and BRV, alongside the ability to modulate inflammatory mediators and pathways associated with T helper

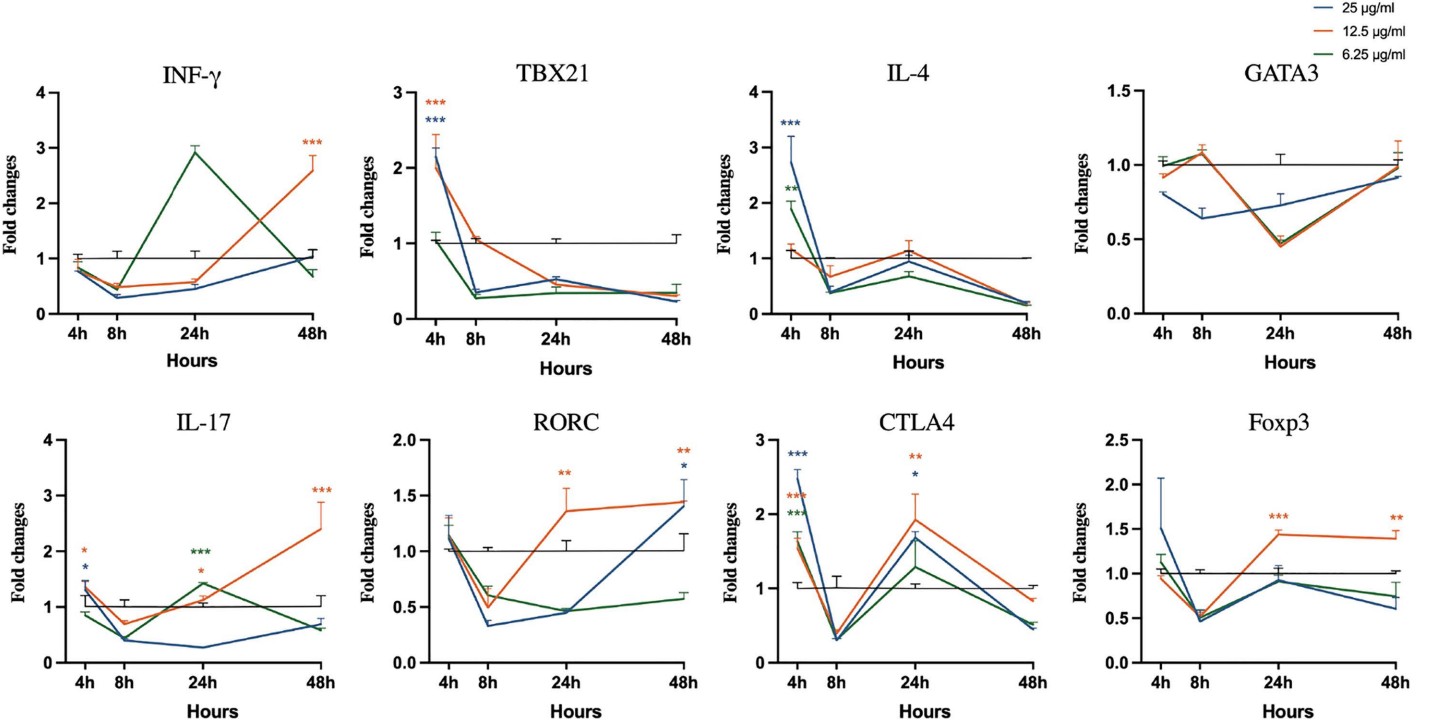

**Fig 9. Analysis of cytokine and transcription factor gene expression in bovine peripheral blood mononuclear cells stimulated with Extract C.** Statistical significance relative to the negative control at each time point was evaluated using one-way ANOVA with Dunnett's post-hoc test (*p < 0.05, **p < 0.01, ***p < 0.001).

cells in RAW264.7 and PBMCs. The observed response patterns, which were dependent on concentration and context, highlight the multicomponent and multitarget potential of these formulations. Collectively, these findings provide a mechanistic basis for further investigation of these compound herbal extracts as potential therapeutic agents for neonatal calf diarrhea.

## Supporting information

**S1 Fig. Analysis of cytokine gene expression in RAW 264.7 cells stimulated with a positive control.** The positive control contained 1 μg/mL lipopolysaccharide derived from *Escherichia coli*. Asterisks (**p < 0.01, ***p < 0.001) indicate significant differences compared with the negative control group at the same time points.
(TIF)

**S2 Fig. Analysis of immunomodulatory gene expression in bovine peripheral blood mononuclear cells stimulated with a positive control.** The positive control contained 1 μg/mL lipopolysaccharide derived from *Escherichia coli*. Asterisks (**p < 0.01, ***p < 0.001) indicate significant differences compared with the negative control group at the same time points.
(TIF)

**S3 Table. Supporting materials for figures.**
(ZIP)

## Acknowledgments

We thank K Pharms Co., Ltd. in Suwon, Korea, for providing Extracts A, B, and C used in this study. We also acknowledge the BK21 and Veterinary Research Institute of Seoul National University for providing access to the laboratory facilities used in this study.

## Author contributions

**Conceptualization:** Han Sang Yoo.

**Data curation:** Xi-Rui Xiang, Eun-Seo Lee, Junho Lee, Han Sang Yoo.

**Formal analysis:** Xi-Rui Xiang, Eun-Seo Lee, Junho Lee, Su Min Kyung.

**Funding acquisition:** Han Sang Yoo.

**Investigation:** Xi-Rui Xiang, Eun-Seo Lee.

**Methodology:** Xi-Rui Xiang, Eun-Seo Lee, Junho Lee, Su Min Kyung.

**Project administration:** Han Sang Yoo.

**Software:** Xi-Rui Xiang, Eun-Seo Lee, Su Min Kyung.

**Supervision:** Han Sang Yoo.

**Validation:** Junho Lee, Su Min Kyung, Han Sang Yoo.

**Visualization:** Eun-Seo Lee, Junho Lee, Su Min Kyung.

**Writing – original draft:** Han Sang Yoo.

**Writing – review & editing:** Han Sang Yoo.

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
