## [Decision Letter · Decision Letter 0]

20 Oct 2025

Dear Dr. Yoo,

Thank you for submitting your manuscript to PLOS ONE. After careful consideration, we feel that it has merit but does not fully meet PLOS ONE’s publication criteria as it currently stands. Therefore, we invite you to submit a revised version of the manuscript that addresses the points raised during the review process.

Please submit a **revised version** of your manuscript addressing the comments below. Please also include a **point-by-point response** explaining how each issue has been resolved.

We look forward to receiving your revised manuscript.

Kind regards,

Tofazzal Md Rakib, PhD

Academic Editor

PLOS ONE

Journal Requirements:

“This study was supported by SMTECH grant number S3346171, the BK21 FOUR program, and the Research Institute for Veterinary Science, Seoul National University, Republic of Korea.”

6. Please include captions for your Supporting Information files at the end of your manuscript, and update any in-text citations to match accordingly. Please see our Supporting Information guidelines for more information: http://journals.plos.org/plosone/s/supporting-information .

Additional Editor Comments:

Thank you for submitting your manuscript to PLOS One. Your paper has now been reviewed, and I am pleased to inform you that reviewers found your study valuable and scientifically sound. However, they have suggested several revisions that need to be addressed before the manuscript can be accepted for publication.

Based on their constructive feedback, I invite you to submit a revised version of your manuscript addressing the comments below. Please also include a point-by-point response explaining how each issue has been resolved.

Reviewer's Responses to Questions

**Comments to the Author**

1. Is the manuscript technically sound, and do the data support the conclusions?

Reviewer #1: Yes

Reviewer #2: Yes

Reviewer #3: Yes

Reviewer #4: Yes

2. Has the statistical analysis been performed appropriately and rigorously?

Reviewer #1: Yes

Reviewer #2: Yes

Reviewer #3: Yes

Reviewer #4: Yes

3. Have the authors made all data underlying the findings in their manuscript fully available?

Reviewer #1: Yes

Reviewer #2: No

Reviewer #3: Yes

Reviewer #4: Yes

4. Is the manuscript presented in an intelligible fashion and written in standard English?

Reviewer #1: Yes

Reviewer #2: Yes

Reviewer #3: Yes

Reviewer #4: Yes

Reviewer #1: The manuscript presents an interesting and valuable study demonstrating the potential of several plant extract formulations to exhibit antiviral activity against BRV and BCoV, as well as their immunomodulatory properties. The article is well written and conducted at a high technical level; however, there are several major and minor comments that need to be addressed.

Minor comments:

Line 25 What is Extract C? Specify the composition.

Line 28 Please use a different word rather than Sophisticated

Line 185 The conclusion of “good biological safety,” which is based solely on the inability of the extracts to suppress the growth of immortalized cells, appears highly presumptive.

Lines 287, 372 and elsewhere in the manuscript: The tested extracts exhibited an antioxidant effect, as evidenced by the suppression of NO production; however, they also showed a pro-inflammatory effect, since some of them increased the levels of pro-inflammatory cytokines IL-6, IFNg and TNF-α. Therefore, the claim of an anti-inflammatory effect should be reconsidered.

Line 288. An evaluation of T cell subsets was not performed in this work, please rephrase

Discussion. The Discussion should focus on interpreting the findings rather than repeating the results or referencing the figures again

Major comments:

Fig. 2 The authors should discuss the discrepancy between the upregulation of iNOS mRNA expression and the significant reduction in NO production observed following treatment with Extract C.

Line 238: FOXP3 is considered a T cell specific marker, particularly of Tregs, it is unlikely that RAW264.7 express FOXP3 mRNA under normal conditions. Please check the specificity of your primers. Wouldn’t it be more appropriate to assess FoxP3 expression following extract treatment in PBMC fraction?

Reviewer #2: The current manuscript, entitled “Antiviral, anti-inflammatory and immunomodulatory effects of herbal extracts of Boswellia serrata, Commiphora myrrha, Scutellaria baicalensis, Gardenia jasminoides, Nypa fruticans and propolis,” discusses the in vitro antiviral, anti-inflammatory, and the immune modulation effects of three different herbal combinations. Although the topic is interesting, the manuscript has some concerns that need to be addressed before it can be considered for publication.

Major concerns:

Introduction:

- The current study is in vitro but the introduction focused on calf diarrhea rather than discussing the components of the different herbal combinations used in the study. I would recommend rewriting the introduction to focus on the different active compounds in the herbal combinations and the previous studies that support them.

- The objectives are vague and need to be rewritten to reflect the current study objectives and have to clarify that the evaluation will be done in vitro.

- Lines 70-72: “The goal of this study was to identify a safe and effective candidate that offers a natural therapeutic option for viral diarrhea in calves and aligns with the “One Health” concept, which emphasizes maintaining ecosystem balance and human well-being by safeguarding animal health”. With the current study design (in vitro), the authors can’t identify the safety or evaluate the effectiveness against viral diarrhea in calves, which will require an in vivo study, which is not the case for the current study.9

Materials and methods:

- There is no clear timeline for the current study.

- There is no mention of where that work was done (which lab) and who did the lab work.

- Line 156 “Table 2Error! Reference source not found..” what does that mean?

- For the cytotoxicity assay lines 107-121. It is not clear how many wells were assigned for control and each extract.

- Why the PC was only used for the PBMCs in the cytotoxicity testing?

- For the testing of antiviral effect on BCoV and BRV lines 129-141, it is not clear how many cells were assigned for controls and each extract.

- For testing of anti-inflammatory effects onRAW264.7 cells lines, 145-156, it is not clear how many cells were assigned for controls and each extract.

- For testing of Immunomodulatory effects on RAW 264.7 cells and bovine PBMCs, lines 160-167, it is not clear how many cells were assigned for controls and each extract.

- For the primers that referenced as this study, there is no explanation of how the authors decided or designed these primers. Please add a section explaining the procedures that were followed by the authors to design these primers.

- Statistical analysis, Was there any pairwise comparisons done? If yes what are they? If not why not?

Results:

- Line 182: “1Error! Reference source not found.,” what does that mean?

- The colors for figures 4-6 are not clear, I would recommend using colors that can be easily distinguished.

Discussion:

Lines 288-290: “Collectively, these findings support a new comprehensive treatment strategy for calf viral diarrhea through the synergistic effect of multiple targets and provide new ideas for prevention and treatment.” That statement is overestimated the study outcomes, please be more specific to the study outcomes as the current study is only in vitro study.

Reviewer #3: General: The authors conduct a commendable study on a wide range of cellular inflammatory responses to three largely composite plant extracts.

Specific:

1. Although the cellular responses to these extracts are quite notable, there are some issues surrounding the extracts themselves that could be better clarified.

First, it would be helpful if the authors could provide a reference as well as some sort of explanation as to how these extracts were prepared in the Herbal Materials section. It would also be good to know why these Extracts were formulated as they are, &/or why these particular extracts were selected for study.

Second, since Extract C is apparently the same as Extract A but with 2 additional components, it would be extremely valuable to directly compare any differences observed between the responses elicited with & without these 2 components. My apologies if any such comparisons in this regard were missed.

Lastly, the manuscript's title may also be somewhat misleading in this respect. It may imply to readers that extracts from each of these sources are being assessed individually. Something less specific may be better, such as; 'Anti-viral, anti-inflammatory, and immunomodulatory effects of herbal extracts in a cellular model of bovine diarrhea', although not all cells were bovine so this may not be a perfect option.

2. Correct referencing errors flagged on lines 156 & 182.

Reviewer #4: I commend the authors for a high quality scientific research. All sections were well written and easy to understand. The results were detailed and the visualization appropriate. The conclusion matched the stated objectives of the study. It is a very good manuscript with a high potential of contributing the advancement of animal and public health.

**Do you want your identity to be public for this peer review?** For information about this choice, including consent withdrawal, please see our Privacy Policy

Reviewer #1: **Yes:** Perfilyeva V. Yuliya

Reviewer #2: No

Reviewer #3: No

Reviewer #4: No

---

## [Author Response · Author response to Decision Letter 1]

30 Nov 2025

Response Letter to Reviewers' Comments

Dear Editor and Reviewers,

We are pleased to resubmit a comprehensively revised and substantially improved manuscript for possible publication in PLOS ONE.

Below we present our detailed responses to each comment from the editor and reviewers. We appreciate the insightful and constructive comments, which have helped us to further improve the manuscript. Each comment has been addressed carefully, and the corresponding revision have been incorporated into the text.

To facilitate your review of our responses and the revised manuscript, we have used the following color-coding approach in our reply letter.

Black text indicates document navigational headings and identifiers.

Grey text represents the original comments from the editor and reviewers.

Purple text indicates our direct responses to the editor's and reviewers' comments.

Additionally, we have prepared a comprehensive table of contents to aid navigation and enable quick access to specific sections.

Table of Contents

JOURNAL REQUIREMENTS AND RESPONSE: 2

REVIEWER #1 COMMENTS AND RESPONSE: 3

REVIEWER #2 COMMENTS AND RESPONSE: 6

REVIEWER #3 COMMENTS AND RESPONSE: 11

REVIEWER #4 RESPONSE: 13

Journal Requirements and Response:

Response:

Thank you for pointing this out. We have carefully reviewed the PLOS ONE manuscript preparation and file naming guidelines and revised all submitted files accordingly.

2. Please state what role the funders took in the study.

3. Please note that funding information should not appear in any section or other areas of your manuscript.

Response

We have updated the funding information as required.

4. When completing the data availability statement of the submission form, you indicated that you will make your data available on acceptance. We strongly recommend all authors decide on a data sharing plan before acceptance, as the process can be lengthy and hold up publication timelines.

Response

We confirm that all underlying data will be made fully available upon acceptance.

5. Your ethics statement should only appear in the Methods section of your manuscript.

6. Please include captions for your Supporting Information files at the end of your manuscript, and update any in-text citations to match accordingly.

Response

We have reviewed the manuscript as required.

7. Please review your reference list to ensure that it is complete and correct.

Response

We have checked the reference list and confirmed that no retracted papers are cited. Four new references were added (now listed as 36, 37, 65 and 66), and the previous reference 57 has been moved to 41 following revisions to the Introduction.

Reviewer #1 Comments and Response:

1. Line 25 What is Extract C? Specify the composition.

Response:

Thank you for your comment. To avoid confusion in the Abstract, we have revised the wording to clarify that the three formulations used in this study (lines 15-17) are 'three distinct herbal extracts (designated Extracts A, B, and C)'. This explicitly defines Extract C as requested.

2. Line 28 Please use a different word rather than Sophisticated

Response:

We agree and have replaced 'sophisticated' with 'complex' in the revised manuscript, as requested (line 24).

3. Line 185 The conclusion of'good biological safety,' which is based solely on the inability of the extracts to suppress the growth of immortalized cells, appears highly presumptive.

Response:

We agree that the term 'good biological safety' is not fully supported by cell viability data alone. Therefore, we have revised the wording to 'did not show obvious cytotoxicity' to more accurately reflect the results (lines 208-209).

4. Lines 287, 372 and elsewhere in the manuscript: The tested extracts exhibited an antioxidant effect, as evidenced by the suppression of NO production; however, they also showed a pro-inflammatory effect, since some of them increased the levels of pro-inflammatory cytokines IL-6, IFNg and TNF-α. Therefore, the claim of an anti-inflammatory effect should be reconsidered.

Response:

Thank you for this critical suggestion. We agree that our data demonstrates a complex, dual effect and that our original claim of a simple 'anti-inflammatory' effect was doubtful. We have addressed this point by revising the text throughout the manuscript (including the Abstract, Discussion, and Conclusion). We have replaced the term 'anti-inflammatory' with the more precise description of 'complex immunomodulatory effects'.

5. Line 288. An evaluation of T cell subsets was not performed in this work, please rephrase.

Response:

We have rephrased this sentence (and similar instances) to refer to 'T helper cell-associated responses' by revising the text throughout the manuscript.

6. Discussion. The Discussion should focus on interpreting the findings rather than repeating the results or referencing the figures again

Response:

Thank you for the comment. We have revised the Discussion to focus on interpretation as reviewer recommended.

7. Fig. 2 The authors should discuss the discrepancy between the upregulation of iNOS mRNA expression and the significant reduction in NO production observed following treatment with Extract C.

Response:

Thank you for this insightful comment. We agree that the divergence between the upregulation of iNOS mRNA and the reduction in NO production observed with Extract C requires clarification.

We have expanded the Discussion to address this point. As now noted in the revised text, iNOS transcription does not always correlate with enzymatic activity or NO output, and several studies have reported similar dissociations in natural-product models. Extract C may suppress NO production through post-transcriptional regulation such as reduced translation or enhanced protein turnover) or by interfering with iNOS catalytic activity, rather than blocking iNOS gene induction. We also have added relevant references (Hung et al., 2019; Tanemoto et al., 2015) to support this interpretation.

8. Line 238: FOXP3 is considered a T cell specific marker, particularly of Tregs, it is unlikely that RAW264.7 express FOXP3 mRNA under normal conditions. Please check the specificity of your primers. Wouldn't it be more appropriate to assess FoxP3 expression following extract treatment in PBMC fraction?

Response:

Thank you for this important comment. We agree that FOXP3 is a marker associated with regulatory T cells and is not expected to be expressed in RAW264.7 macrophages. In line with your suggestion, we have removed FOXP3 data for RAW264.7 cells from the figures and the text. The analysis of FOXP3 is now limited to PBMC samples, where T cell related responses can be evaluated in an appropriate biological context. We also rechecked the FOXP3 primers and confirmed that they produce a single specific product in PBMC samples (Table 3).

Reference

Hung YL, Wang SC, Suzuki K, Fang SH, Chen CS, Cheng WC, et al. Bavachin attenuates LPS-induced inflammatory response and inhibits the activation of NLRP3 inflammasome in macrophages. Phytomedicine. 2019;59:152785. Epub 20181210. doi: 10.1016/j.phymed.2018.12.008. PubMed PMID: 31009850.

Tanemoto R, Okuyama T, Matsuo H, Okumura T, Ikeya Y, Nishizawa M. The constituents of licorice (Glycyrrhiza uralensis) differentially suppress nitric oxide production in interleukin-1β-treated hepatocytes. Biochemistry and biophysics reports. 2015;2:153-9. doi: 10.1016/j.bbrep.2015.06.004.

Reviewer #2 Comments and Response:

Introduction:

1. The current study is in vitro but the introduction focused on calf diarrhea rather than discussing the components of the different herbal combinations used in the study. I would recommend rewriting the introduction to focus on the different active compounds in the herbal combinations and the previous studies that support them.

Response:

Thank you for the helpful suggestion. The Introduction has been reorganized to focus more directly on the active herbal components and the supporting literature (lines 61-72).

2. The objectives are vague and need to be rewritten to reflect the current study objectives and have to clarify that the evaluation will be done in vitro.

Response:

In line with this revision, we have also rewritten the study objectives so that they are more precise and clearly indicate that all evaluations were performed in vitro. The revised objectives appear in lines (lines 73-76).

3. Lines 70-72:'The goal of this study was to identify a safe and effective candidate that offers a natural therapeutic option for viral diarrhea in calves and aligns with the 'One Health'concept, which emphasizes maintaining ecosystem balance and human well-being by safeguarding animal health'. With the current study design (in vitro), the authors can't identify the safety or evaluate the effectiveness against viral diarrhea in calves, which will require an in vivo study, which is not the case for the current study.

Response:

We agree that the original sentence went beyond what can be supported by an in vitro study. Since our work does not include any in vivo experiments, it cannot address safety or clinical effectiveness in calves. To avoid giving an impression that exceeds the scope of the study, we have removed the sentence in lines 70–72 and revised the objectives so that they clearly reflect the in vitro nature of the work (lines73-76). We appreciate the reviewer's careful reading.

Materials and methods

4. There is no clear timeline for the current study.

Response:

We have added the specific experimental time points to clarify the study timeline of each treatment.

5. There is no mention of where that work was done (which lab) and who did the lab work.

Response:

Thank you for the comment. We have added a statement in the Materials and Methods section specifying where the laboratory work was carried out and by whom (lines 80-81).

6. Line 156 'Table 2Error! Reference source not found..' what does that mean?

Response:

We sincerely apologize for this formatting oversight. This text appeared due to a broken cross-reference field in the manuscript file during the conversion process.

7. For the cytotoxicity assay lines 107-121. It is not clear how many wells were assigned for control and each extract.

Response:

We have added the number of replicate wells used for each control and treatment group; each condition was tested in four technical replicates (line 122).

8. Why the PC was only used for the PBMCs in the cytotoxicity testing?

Response:

Thank you for the comment. In RAW 264.7 cells, 1 μg/mL LPS is a widely used standard concentration for macrophage activation and is well established in the literature as non-cytotoxic(Facchin et al., 2022). Because this dose functions as an inflammatory inducer rather than a viability-reducing stimulus, including it as a separate condition in the cytotoxicity assay would not provide meaningful information.

In contrast, LPS is used less frequently in bovine PBMC systems, and in this study an LPS-treated group was required as a positive reference for the downstream immunoactivity assays. For this reason, we confirmed its safety in PBMCs by including it in the viability test, whereas a separate LPS condition was not needed in the RAW 264.7 cytotoxicity assay.

9. For the testing of antiviral effect on BCoV and BRV lines 129-141, it is not clear how many cells were assigned for controls and each extract.

Response:

We have added the number of wells used per condition and confirmed that the assay was performed in three independent repeats (lines 145, 160).

10. For testing of anti-inflammatory effects onRAW264.7 cells lines, 145-156, it is not clear how many cells were assigned for controls and each extract.

Response:

The Methods now specify the wells assigned to each group, and all assays were carried out in three independent experiments (lines 165, 177).

11. For testing of Immunomodulatory effects on RAW 264.7 cells and bovine PBMCs, lines 160-167, it is not clear how many cells were assigned for controls and each extract.

Response:

We have clarified the well numbers for each condition, and these assays were also repeated independently three times (lines 182, 189).

12. For the primers that referenced as this study, there is no explanation of how the authors decided or designed these primers. Please add a section explaining the procedures that were followed by the authors to design these primers.

Response:

Thank you for this suggestion. We have added a detailed description of the primer design procedure in the Methods section (lines 154-157). We clarified that the primers were designed using NCBI Primer-BLAST based on GenBank sequences, and their specificity was validated both in silico (BLAST) and experimentally (melting curve analysis). Please refer to the revised text for details.

13. Statistical analysis, Was there any pairwise comparisons done? If yes what are they? If not why not?

Response:

Yes, pairwise comparisons were performed. Statistical significance was evaluated using one-way ANOVA followed by Dunnett's post-hoc test, comparing each extract-treated group with the corresponding control group. This information has been added to the Statistical analysis section (lines 198-199) and clarified in the figure legends.

Results

14. Line 182: '1Error! Reference source not found.,' what does that mean?

Response:

We sincerely apologize for this formatting oversight. We have corrected the formatting error in the revised manuscript.

15. The colors for figures 4-6 are not clear, I would recommend using colors that can be easily distinguished.

Response:

Thank you for the comment. Figures 4-6 have been updated with clearer and more distinguishable colors.

Discussion

16. Lines 288-290: 'Collectively, these findings support a new comprehensive treatment strategy for calf viral diarrhea through the synergistic effect of multiple targets and provide new ideas for prevention and treatment.' That statement is overestimated the study outcomes, please be more specific to the study outcomes as the current study is only in vitro study.

Response:

We have removed the overstated sentence from the original version and revised the text so that the conclusions and objective reflect only the scope of the in vitro findings.

Reference

Facchin BM, dos Reis GO, Vieira GN, Mohr ETB, da Rosa JS, Kretzer IF, et al. Inflammatory biomarkers on an LPS-induced RAW 264.7 cell model: a systematic review and meta-analysis. Inflammation Research. 2022;71(7):741-58. doi: 10.1007/s00011-022-01584-0.

Reviewer #3 Comments and Response:

1. First, it would be helpful if the authors could provide a reference as well as some sort of explanation as to how these extracts were prepared in the Herbal Materials section. It would also be good to know why these Extracts were formulated as they are, &/or why these particular extracts were selected for study.

Response:

Thank you for this comment. We have revised the Herbal Materials section to include additional information on the source, composition, extraction description, and selection rationale of the three extracts. A brief explanation has been added noting that the extracts were standardized products provided by K Pharms Co., Ltd., produced using standard solvent extraction procedures, and selected based on their traditional use in gastrointestinal or inflammatory conditions (lines 87-91).

2. Second, since Extract C is apparently the same as Extract A but with 2 additional components, it would be extremely valuable to directly compare any differences observed between the responses elicited with & without these 2 components. My apologies if any such comparisons in this regard were missed.

Response:

We thank the reviewer for this insightful comment. The present study was designed to evaluate each extract relative to its control group; therefore, no statistical comparison was performed between Extract A and Extract C. The discussion referring to compositional differences was intended only as a mechanistic hypothesis rather than a direct comparison between extracts.

3. Lastly, the manuscript's titl

---

## [Decision Letter · Decision Letter 1]

10 Dec 2025

Dear Dr. Yoo,

We look forward to receiving your revised manuscript.

Kind regards,

Tofazzal Md Rakib, PhD

Academic Editor

PLOS One

**Journal Requirements:**

**Additional Editor Comments:**

Thank you for submitting the revised version of your manuscript. I want to request you to address the comments/questions raised by Reviewer 3.

Reviewers' comments:

Reviewer's Responses to Questions

**Comments to the Author**

Reviewer #1: All comments have been addressed

Reviewer #3: (No Response)

2. Is the manuscript technically sound, and do the data support the conclusions?

Reviewer #1: Yes

Reviewer #3: Partly

3. Has the statistical analysis been performed appropriately and rigorously?

Reviewer #1: Yes

Reviewer #3: Yes

4. Have the authors made all data underlying the findings in their manuscript fully available?

Reviewer #1: Yes

Reviewer #3: Yes

5. Is the manuscript presented in an intelligible fashion and written in standard English?

Reviewer #1: Yes

Reviewer #3: Yes

Reviewer #1: The manuscript has undergone substantial improvement and now presents strong, publishable findings.

Minor comments:

Line 320 B. serrata

Reviewer #3: General Opinion: The authors have not really addressed any of the initial recommendations adequately.

Specific Recommendations:

1. The additional information provided in the Herbal Materials section is somewhat helpful in explaining the origin of the 3 herbal extracts under study here. However, references are still essential to support "their demonstrated bioactivities in gastrointestinal or inflammatory conditions". Likewise, how are these Extracts currently used commercially? In fact, an internet search for 'K Pharms Co., Ltd. of Suwon, Korea' found no company of that name manufacturing herbal extracts.

2. The request for a direct comparison of Extracts A & C would not be difficult, since the data is already included. In fact, this comparison could actually enhance the paper's "mechanistic hypothesis". In a cursory comparison of the results, it appears that Extracts A & C act very similarly except for the data shown in Fig. 2. This may be an important mechanistic finding, based on the 2 additional components of Extract C.

3. While the manuscript's title is improved without listing the individual extract components, it still doesn't address the potential application of this study. Are these Extracts currently used in animal disease control? Isn't that why bovine PBMC's were employed? Including more information on the background & current usage of these Extracts in the Materials & Methods, as requested above, may help formulate a better title to promote their veterinary usage.

**Do you want your identity to be public for this peer review?** For information about this choice, including consent withdrawal, please see our Privacy Policy

Reviewer #1: **Yes:** Perfilyeva V. Yuliya, PhD, Assoc. Prof.

Reviewer #3: No

---

## [Author Response · Author response to Decision Letter 2]

29 Dec 2025

Response Letter to Reviewers' Comments

Dear Editor and Reviewers,

Thank you very much for your continued guidance. We have carefully addressed the additional comments from the latest review round.

Specifically, we have updated the manuscript title to highlight the clinical application, clarified the manufacturer details in the Methods section, and refined the Discussion to provide better mechanistic insights.

Our detailed responses follow below.

Black text indicates document navigational headings and identifiers.

Blue text represents the original comments from the editor and reviewers.

Purple text indicates our direct responses to the editor's and reviewers' comments.

Additionally, we have prepared a comprehensive table of contents to aid navigation and enable quick access to specific sections.

Table of Contents

REVIEWER #1 COMMENTS AND RESPONSE: 2

REVIEWER #3 COMMENTS AND RESPONSE: 3

Reviewer #1 Comments and Response:

1. Line 320 B. serrata

Response:

Thank you for the careful observation. We have corrected the typographical error in 'B. serrata' as suggested.

Reviewer #3 Comments and Response:

1. The additional information provided in the Herbal Materials section is somewhat helpful in explaining the origin of the 3 herbal extracts under study here. However, references are still essential to support "their demonstrated bioactivities in gastrointestinal or inflammatory conditions". Likewise, how are these Extracts currently used commercially? In fact, an internet search for 'K Pharms Co., Ltd. of Suwon, Korea' found no company of that name manufacturing herbal extracts.

Response:

We appreciate the verification effort. We would like to clarify that the manufacturer, K Pharms Co., Ltd., was formerly known as Yeskin Co., Ltd. and officially changed its name in 2024. This likely explains the difficulty in the internet search. Regarding the application, we have updated the text to indicate that these extracts are formulated for veterinary health support, and we have added representative citations to the Materials section to support the bioactivity claims(line87-89). Furthermore, the manufacturer is still trying to develop as a therapeutics in calf diarrhea. After licensing from Korean government, these materials will be commercialized.

2. The request for a direct comparison of Extracts A & C would not be difficult, since the data is already included. In fact, this comparison could actually enhance the paper's "mechanistic hypothesis". In a cursory comparison of the results, it appears that Extracts A & C act very similarly except for the data shown in Fig. 2. This may be an important mechanistic finding, based on the 2 additional components of Extract C.

Response:

We agree that a direct comparison between Extract A and Extract C provides valuable context for the mechanistic hypothesis. The distinct antiviral profiles observed in Fig. 2 offer supporting evidence for the functional impact of the differential formulation composition. We have updated the Discussion section (lines 324-325) to explicitly incorporate this comparative analysis.

3. While the manuscript's title is improved without listing the individual extract components, it still doesn't address the potential application of this study. Are these Extracts currently used in animal disease control? Isn't that why bovine PBMC's were employed? Including more information on the background & current usage of these Extracts in the Materials & Methods, as requested above, may help formulate a better title to promote their veterinary usage.

Response:

We appreciate this constructive suggestion. We have revised the manuscript title to 'Antiviral and Anti-inflammatory Evaluation of Herbal Extracts: Implications for the Management of Calf Diarrheal Diseases' to explicitly highlight the clinical application. This update clarifies the veterinary context and justifies the use of bovine specific models. We tried to reveal the immunomodulatory effects of those materials in bovine. Although we tried to find out appropriate bovine immune cells to conduct our study, we could not find the cell line. Therefore, we decided to use bovine PBMC.

We thank the reviewers for their constructive feedback, which has significantly improved the quality of our work. We hope the revised manuscript is now suitable for acceptance.

Sincerely,

The authors

---

## [Decision Letter · Decision Letter 2]

15 Jan 2026

Antiviral and Anti-inflammatory Evaluation of Herbal Extracts: Implications for the Management of Calf Diarrheal Diseases

PONE-D-25-49741R2

Dear Dr. Yoo,

We’re pleased to inform you that your manuscript has been judged scientifically suitable for publication and will be formally accepted for publication once it meets all outstanding technical requirements.

Kind regards,

Tofazzal Md Rakib, PhD

Academic Editor

PLOS One

Additional Editor Comments (optional):

Thank you for submitting the revised version of your manuscript. The reviewer#3 has raised a minor comment; consider addressing it in the final version.

Reviewers' comments:

Reviewer's Responses to Questions

**Comments to the Author**

Reviewer #3: (No Response)

2. Is the manuscript technically sound, and do the data support the conclusions?

Reviewer #3: Yes

3. Has the statistical analysis been performed appropriately and rigorously?

Reviewer #3: Yes

4. Have the authors made all data underlying the findings in their manuscript fully available?

Reviewer #3: Yes

5. Is the manuscript presented in an intelligible fashion and written in standard English?

Reviewer #3: Yes

Reviewer #3: The authors addressed the latest comments on their interesting study fairly well, which has much improved their manuscript. The only remaining minor recommendation is that it would be best to cite the Yeskin Co. parenthetically after K Pharms in line 84 of the Herbal Materials section; i.e. ....by K Pharms Co., Ltd. (Suwon, Korea; formerly Yeskin Co., Ltd.). Yeskin still retains a much more accessible internet profile, which will strengthen the study of their extracts.

**Do you want your identity to be public for this peer review?** For information about this choice, including consent withdrawal, please see our Privacy Policy

Reviewer #3: No

---

## [Editor Report · Acceptance letter]

PONE-D-25-49741R2

PLOS One

Dear Dr. Yoo,

I'm pleased to inform you that your manuscript has been deemed suitable for publication in PLOS One. Congratulations! Your manuscript is now being handed over to our production team.

Kind regards,

on behalf of

Dr. Tofazzal Md Rakib

Academic Editor

PLOS One